# Decomposing the Spatial and Temporal Effects of Climate and Habitat on a Hazel Grouse (*Tetrastes bonasia*) Population in Northeastern Chinese Mountains

**DOI:** 10.3390/ani13122025

**Published:** 2023-06-18

**Authors:** Xiaoying Xing, Yuesen Zhang, Xiang Li, Guangshun Jiang

**Affiliations:** 1College of Wildlife and Protected Area, Northeast Forestry University, Harbin 150040, China; ab71588@163.com (X.X.);; 2Feline Research Center of National Forestry and Grassland Administration, College of Wildlife and Protected Area, Northeast Forestry University, Harbin 150040, China; 3Northeast Asia Biodiversity Research Center, Harbin 150040, China

**Keywords:** hazel grouse, camera trapping, road, human disturbance, human activity

## Abstract

**Simple Summary:**

The timely monitoring of the population fluctuations of endangered species and discovering their causes are critical for biodiversity conservation in mountainous areas. To monitor population dynamics and explore the effects of climate and habitat on the population distribution of the hazel grouse, a second-class protected animal in China, infrared cameras were installed in Hunchun, China (Jilin Province). The hazel grouse preferred stable climate conditions. A distribution close to paved roads in the summer benefitted birds’ survival and breeding, but the activity of local people in the mountain disturbed them significantly in autumn. We report here how the hazel grouse has responded to anthropogenic disturbances in the mountains of northeast China over a decade, and we call for further attention to this species that is sensitive to climatic fluctuations at high latitudes.

**Abstract:**

Habitat, climate, and human disturbances have important effects on wildlife, and these are especially critical for threatened species. In this study, we used infrared camera traps to monitor the population dynamics of the hazel grouse (*Tetrastes bonasia*) from 2012 to 2021 in northeast China and explore the effects of habitat, climate, and human disturbance on their distribution. We analyzed 16 environmental variables related to significant differences between presence recordings and absence recordings within and between seasons. Temperatures and roads influenced the distribution of the hazel grouse, but topography and vegetation types did not. The hazel grouse preferred deciduous forest and oak forest from spring to autumn. This study provides ecological information to help guide the mountain habitat management of the hazel grouse in national parks.

## 1. Introduction

The hazel grouse (*Tetrastes bonasia*) is a bird that typically occurs in mixed coniferous deciduous woods at high latitudes and is distributed broadly in northern Eurasia [1]. In northeast China, the species is distributed in many different types of forest habitats. It is listed as vulnerable species in the European Union [2], and has been listed as a national second-class protected animal in China [3]. Although it is still classified as a species of least concern by the International Union for Conservation of Nature [1], population numbers have had a decreasing trend. A main reason for this is the high degree of overlap between the hazel grouse distribution and areas of human activity, which leads to habitat loss at local and regional scales [4]. However, how anthropogenic and ecological factors interactively influence the spatial and temporal distributions of the hazel grouse remains unclear.

Understanding the spatial distribution dynamics of hazel grouse populations and uncovering how to use environmental resources appropriately to maximize their fitness are critical to biodiversity conservation and monitoring habitat quality alterations in temperate zones [5,6,7]. However, little is known about the ecological needs and spatial distribution dynamics of the hazel grouse because of the difficulty in collecting monitoring data in the field [8,9]. Various selection pressures work together to affect the use of various resources in their habitats and, thus, affect spatial distribution [10,11,12]. For example, habitat, climate, and human disturbances all play important roles in the distribution, diversity, and abundance of species. Unraveling the interactions between these factors and population dynamics is crucial for species conservation and management [13].

Geographical factors affect species distributions and resource utilization [12,14]. The hazel grouse is typically distributed in mountains and forests in temperate zones. Geographical parameters such as slope, orientation, and elevation can directly or indirectly affect their spatial distribution and population fluctuations by influencing vegetation types and the local microclimate. The hazel grouse in Korea has an obvious seasonal altitudinal gradient movement, and is distributed mainly at 600–800 m elevation in the winter and at 800–1000 m in the spring [15]. In the Changbai Mountains of China, the hazel grouse is mainly distributed at 900–1500 m elevation and moves to lower altitudes in the winter [16].

Global climate change significantly affects the distribution of species, especially in high-latitude areas [17,18,19]. Assessing the extent to which population dynamics respond to climate change is critical for species management and biodiversity conservation, especially for species living in high-latitude temperate regions. Climate can affect the dynamics and distribution of bird communities [20,21]. In some areas, climate change can affect bird community structures even more than habitat changes [22]. For example, low temperatures, below thermal-neutral temperatures, limit the distribution of birds in cold regions by increasing the energy costs of survival [23,24]. Research on other grouses, such as the ruffed grouse (*Bonasa umbellus*) and black grouse (*Lyrurus tetrix*), has shown negative correlations between precipitation during the breeding season and breeding success and population growth [25,26]. In these species, chicks are very sensitive to cold and humidity, and high precipitation levels affect their thermoregulation and reduce foraging time, food availability, and, thus, the survival of broods [27,28]. In contrast, the thicker winter snow cover provides the grouse with deeper, drier snow burrows [29], and burrowing in snow reduces heat loss and at the same time hides them from predators [30].

Human disturbances are critical factors influencing population distributions and species diversity [31,32,33]. Nature reserves and national parks have been established in the mountainous areas of northeastern China in recent years, which have largely relieved the destruction of forest habitats by human activity. In 2021, the Northeast China Tiger and Leopard National Park (Dongning, China) was officially established to effectively protect and restore wild populations of the Siberian tiger (*Panthera tigrisaltaica*) and Amur leopard (*Panthera pardus orientalis*) and to ensure their stable reproduction and survival in China. Human disturbances, such as grazing, gathering herbs, and digging for wild vegetables, have since decreased in the national park areas. As one of the national protected key bird species in this national park, there is an urgent need to evaluate if and how hazel grouse distributions change under the environmental alterations caused by different management strategies [34].

Infrared camera traps are widely used for monitoring wild animals and can provide information on population distributions and dynamics under habitat or climate changes, as well as how wildlife respond to human disturbances [35,36,37]. Field surveys and radio tracking, instead of infrared camera traps, have been commonly used for surveying and monitoring the hazel grouse [8,30,38,39]. However, the hazel grouse is fast-moving and alert when on the ground, making it difficult to directly observe. Researchers have begun using infrared camera traps that can take a series of photographs over long periods in large areas without disturbing the hazel grouse to obtain population distribution dynamics on a large spatiotemporal scale [40,41]. The present study used infrared camera traps to monitor hazel grouse in Hunchun in the Northeast China Tiger and Leopard National Park over a 10-year period, aiming to reveal the combined effects of habitat traits, climate change, and human disturbances on habitat selection and population distribution dynamics. Combined with prior studies, we expected that the hazel grouse prefers (1) distributing in patches dominated by deciduous trees; (2) lower precipitation areas in the breeding season; and (3) areas with less human interference. The results of this study will provide guidance for policy making regarding grouse conservation and management, biodiversity conservation, and ecosystem restoration in national parks.

## 2. Materials and Methods

### 2.1. Study Site and Infrared Camera Trap Installation

The core area of the Northeast China Tiger and Leopard National Park is located in Hunchun, Jilin Province, China. It covers the area from 130°14′8″ E–131°14′44″ E longitude and 42°32′40″ N–43°28′0″ N latitude, with a total area of 108,700 km^2^ at an altitude of 5–973 m. This area has a medium-latitude monsoon climate, with an average annual precipitation of 618 mm and an average annual temperature of 5.6 °C. The climate varies greatly between seasons. Forest is the dominating landscape type, covering 74% of the reserve, while farmland accounts for 8.1% of the area. The vegetation type is mainly deciduous forest and oak forest, with a small amount of coniferous forest and mixed coniferous forest in the north.

A grid of 2 km × 2 km squares was used as the working unit. The cameras used were Ltl6210mc (Ltl Acorn, Zhuhai, China), L710-940 (Yianws, Shenzhen, China), and UVL4 (UOVISION, Shenzhen, China). These cameras can record photographs and videos in visible light during the day and in infrared at night. One camera was set in each grid space, with 165 cameras in total covering all different elevations and habitat types within the reserve. Infrared cameras were placed at least 250 m from each other [42]. The infrared cameras were tied to vertical tree trunks of moderate thickness. The camera view and shooting angle were estimated and adjusted, and the appropriate lens direction was selected to obtain the best possible view. Obstacles and plants near the camera were properly cleared to avoid unnecessary triggering of the camera [43,44]. The latitude and longitude were measured using GPS, and cameras were numbered. The shooting mode of the infrared cameras was set to “shooting + recording” (three photos plus 30 s videos after triggering). The triggering interval was 30 s, and the sensitivity was set to medium. The battery and memory card were replaced every 2 months between January 2012 and December 2021.

### 2.2. Data Extraction and Processing

Information on the presence of animals at each camera was extracted, including location, time, species, and number of valid photos. Continuous photos or videos of the same individual at the same camera site within 30 min were counted as one independent valid photo [45]. We considered it to be the same individual because it kept appearing in a continuous series of photos or videos and moving continuously in space. For the cameras that captured the hazel grouse, we defined each valid photo of the hazel grouse as a presence recording. This means the same cameras recorded multiple presence recordings, but the recordings of the same camera were from different times and dates. Additionally, we defined presence as the detection of a grouse by the trail cameras. For the cameras that did not capture the hazel grouse but captured other birds, the same method was used, and each valid photo of a birdwas recorded as an absence recording. Once we obtained a valid photo of a bird, we collected the data according to the time and place of its occurrence and classified it according to the season of its presence, with data for each season being independent. For example, if a particular camera did not record a hazel grouse throughout the spring over all the 10-year period, then it was considered that no hazel grouse was present at that camera site in the spring, and the recordings of other birds it captured were treated as absence recordings for the spring. If, however, the site photographed a hazel grouse in summer, then the valid photo of the hazel grouse can be considered as a summer presence recording, and the other birds photographed by this camera in summer cannot be considered as absence recordings. The number of presence recordings and the number of absence recordings were counted, and we selected a certain number of recordings from the absence recordings with simple random sampling to compare with the presence recordings for analysis.

We selected 16 environmental variables, including topography, vegetation, human disturbances, and climate (Table 1). Topography included slope, orientation, and elevation. The slope, orientation, and elevation were assigned using a digital elevation model of the Hunchun area in Arcgis software (v10.2; Esri, Redlands, CA, USA): north (0–22.5) (337.5–360), northeast (22.5–67.5), east (67.5–112.5), southeast (112.5–157.5), south (157.5–202.5), southwest (202.5–247.5), west (247.5–292.5), and northwest (292.5–337.5). The digital elevation model map was downloaded from the International Scientific Data Service website “https://www.casdc.cn/” (accessed on 20 December 2022). Data on vegetation included the normalized difference vegetation index (NDVI) and vegetation types, among which the land-use classes were divided into nine types: mixed forest, Korean fir forest, deciduous forest, oak forest, larch forest, spruce forest, birch, farmland, and village. Climate data were downloaded from WorldClim “https://www.worldclim.org/data/worldclim21.html” (accessed on 20 December 2022), and all were annual values. Considering different vegetation structures among seasons, we classified the variables according to season.

### 2.3. Statistical Analyses

We used SPSS (v.22.0; IBM Corp., Armonk, NY, USA) and R (v.4.2.2) for data analysis, and all tests were two-tailed. First, we used the Shapiro–Wilk test to check whether the samples were normally distributed, and all samples were not normally distributed. The Mann–Whitney U test was used to explore differences between presence and absence recordings in each season, and the vegetation type between presence and absence recordings in each season was analyzed using the correspondence analysis (R-Q factor analysis). Then, we used the Kruskal–Wallis test with Bonferroni correction and the Mann–Whitney U test to explore whether topography, vegetation, and human disturbances had significant differences between presence recordings from different seasons. At this step, vegetation type was analyzed using correspondence analysis.

## 3. Results

The videos and photos captured by 165 infrared cameras in Hunchun area from 2012 to 2021 were extracted, identified, and collated. A total of 46 infrared camera sites had captured images of hazel grouse, including 286 photos and 30 videos (267 s in total). A total of 101 hazel grouse presence recordings were counted. There were 29 spring presence recordings, 42 summer presence recordings, 28 autumn presence recordings, and 2 winter presence recordings. Because there were only two winter presence recordings, we did not analyze variables of presence and absence recordings in winter. As for absence recordings, 107 were chosen for analysis, with 31 spring absence recordings, 45 summer absence recordings, and 31 autumn absence recordings.

### 3.1. Topography and Vegetation

There were neither significant differences in topography and vegetation variables between presence and absence recordings in each season (Table 2, Figure 1) nor in presence recordings among three different seasons (winter excluded; Table 3, Figure 1). As for vegetation type, the hazel grouse was mainly distributed in deciduous forest, oak forest, mixed forest, and Korean fir forest (Figure 2).

### 3.2. Climate

In summer, compared with absence recordings, presence recordings had significantly higher annual mean temperatures (*t* = −2.007, *p* = 0.045; Table 2, Figure 1e) and significantly smaller annual temperature ranges (*t* = 2.654, *p* = 0.008; Table 2, Figure 1g).

### 3.3. Human Disturbance

In summer, the distance to paved road for presence recordings was significantly lower than for absence recordings (*t* = 2.142, *p* = 0.032; Table 2, Figure 1l). For all presence recordings, the distance to unpaved road in spring (*t* = −2.608, *p* = 0.027; Table 3, Figure 1k) and summer (*t* = −2.575, *p* = 0.030; Table 3, Figure 1k) were significantly lower than in autumn.

## 4. Discussion

There were some differences between hazel grouse presence and absence recordings in summer, and there were also some differences between presence recordings in different seasons. Hazel grouse were rarely photographed in winter, mainly because they hide in snow burrows in winter [46].

### 4.1. Topography and Vegetation Did Not Affect Hazel Grouse Population Distribution

Contrary to our expectations, topography and vegetation had no significant effects on the distribution of the hazel grouse. This may be due to the relatively homogeneous topography and vegetation types in the national park; the national park is mainly dominated by low mountains and hills, with a forest coverage rate of 74%, and concentrated in mountain areas, while the plain areas are more commonly exploited by humans, so the hazel grouse are concentrated in the forest-covered mountain areas. The vegetation types are mainly deciduous forest and oak forest, and the distribution is also homogeneous. However, most of cameras were located in the forest, so this may have introduced bias into the data. These results cannot indicate a habitat preference of the hazel grouse and further exploration is needed.

Hazel grouse were mostly active at elevations between 200 and 450 m, which is lower than previously reported from other areas [15,16], likely mainly due to the local geographical characteristics. The landform of the Hunchun area of the Northeast China Tiger and Leopard National Park is mainly low mountains, and the highest point is only 973 m above sea level, which is mostly low hills. Moreover, deciduous forests are distributed at lower altitudes, while coniferous forests are distributed at higher altitudes [47]. This is in line with our hypothesis stating that hazel grouse prefer distributing in patches dominated by deciduous trees.

As for the selection of vegetation types, there were no significant differences among different seasons, and hazel grouse dominantly appeared in deciduous patches and oak patches but were not recorded in larch or spruce patches. These non-significant differences may be caused by the fact that deciduous forest is not only the dominant vegetation type in this area but also that it is the habitat type preferred by hazel grouse. The present results were in line with other studies [48,49] that indicated that deciduous forests can provide hazel grouse with sufficient food and anti-predation places [15]. The oak patches in this area are mainly composed of Mongolian oak (*Quercus mongolica*) in natural secondary deciduous forest after logging or original mixed forest after logging red pine. Deciduous patches and oak patches are rich in inflorescences, buds, and fruits, from which hazel grouse can feed. Moreover, the complex vegetation structure can help hazel grouse hide themselves from predators. However, many studies in other areas have shown that coniferous forest is also an indispensable habitat [50,51,52] that can help hazel grouse avoid predators in winter when the vegetation structure of deciduous forest is reduced, especially when snow cover is not enough for hiding. Because the winter is a critical period for hazel grouse, factors affecting the winter distribution of the hazel grouse need further research.

### 4.2. Climate Effects

Climate factors that have significant influences on bird distribution mainly include temperature and precipitation [53]. In summer, the annual mean temperature of presence recordings was significantly higher than that of absence recordings. This suggested that higher annual mean temperatures are favorable for hazel grouse in the summer. In the pre-laying period of the hazel grouse in the spring, warm weather is conducive to accelerating snow melting, promoting plant growth, and providing nutritious food for females to form viable eggs, promoting breeding success, and this may therefore lead to more presence recordings in the summer breeding period [54,55]. However, the annual temperature range of presence recordings was significantly smaller than that of absence recordings in summer. The annual temperature range is the difference between the annual maximum and minimum temperatures. Therefore, a smaller annual temperature range may indicate that the hazel grouse preferred an environment with a stable temperature in the breeding season. Animals often prefer areas with relatively stable temperatures and little variation in order to keep themselves and/or their offspring within a stable thermoneutral zone as much as possible [56]. Whether or how global temperature changes affect the population and distribution dynamics of the hazel grouse deserves further attention.

### 4.3. Human Disturbances

More hazel grouses appeared significantly closer to paved roads in presence recordings compared with absence recordings in the summer. Compared with autumn, hazel grouse were recorded closer to unpaved roads in the spring and summer. We supposed that this was due to an increase in suitable habitats nearby roads in the summer. Studies around roads in Europe showed that roads have a positive effect on birds of open and semi-open environments, suggesting that roads can provide marginal habitats and hedgerows, reducing predation pressures and providing a warmer area for some birds to breed and feed nestlings [57,58]. However, the above results do not suggest that human activity promotes hazel grouse survival and breeding. On the contrary, the hazel grouse appeared significantly further away from unpaved roads in autumn than in spring and summer, which was in accordance with the fact that local villagers climb the mountains more in autumn to collect mountain products such as mushrooms and edible wild herbs. In May and June, when food is relatively scarce, the hazel grouse may venture to the dirt roads to find food. Some researchers believe that the lack of upper-layer plants and abundant sunlight on unpaved roads in the forest can stimulate the growth of food for the hazel grouse, thus attracting them to forage [38]. However, in autumn, when the grouse can find abundant food in the forest, they can stay away from the unpaved roads where human activity is frequent.

Study have shown that although Asian populations of hazel grouse are also affected by habitat fragmentation, they are less affected by habitat fragmentation than European populations [48]. Possible reasons include that Asian populations adapt to more open habitats, can spread over long distances [59], and form flocks in winter [49,60]. Such characteristics may enable them to move more actively through open habitats to other forest patches. Additionally, although the camera’s location was closed to the road, most of the local roads were adjacent to mountains and the side close to the road was usually steep and smooth, separating the road from the forest like a barrier, and the interference from roads was generally small. However, grouse were recorded less frequently close to unpaved roads in autumn, which suggested that human activity in autumn could lead to negative effects on hazel grouse in Hunchun National Park. Villagers collecting mushrooms walking along the road made various noises that may scare the grouse. The hazel grouse is a national second-class protected animal, which is protected from hunting under strict laws in China; however, human activity and disturbances still exist, even in the core of nature reserves. Through experiments and interactions with local villagers, we found that many human activities still exist deep in the reserve, such as cattle grazing and farmers collecting mushrooms and wild vegetables. How these human disturbances, combined with landscape parameters [48,61], affect breeding and population fluctuation needs to be investigated further.

## 5. Conclusions

The present study showed no influences of topography and vegetation on hazel grouse, while temperature and human disturbance can affect the population distribution of hazel grouse in the Northeast China Tiger and Leopard National Park in Hunchun, China. The distribution of hazel grouse in the park was affected by temperature and roads, especially in summer, indicating that the protection of the hazel grouse in the national park should focus on areas near paved roads in summer, such as putting signs along roads to alert people of the presence of grouse. Further, the present data demonstrated a necessity for limiting access to the mountains in autumn. Understanding the habitat and climate requirements of endangered species is important for species conservation, and the effect of human disturbance on them also needs to be clarified. Globally, we need to know more about the habitats where endangered species live to protect them and prevent them from extinction.

## Figures and Tables

**Figure 1 animals-13-02025-f001:**
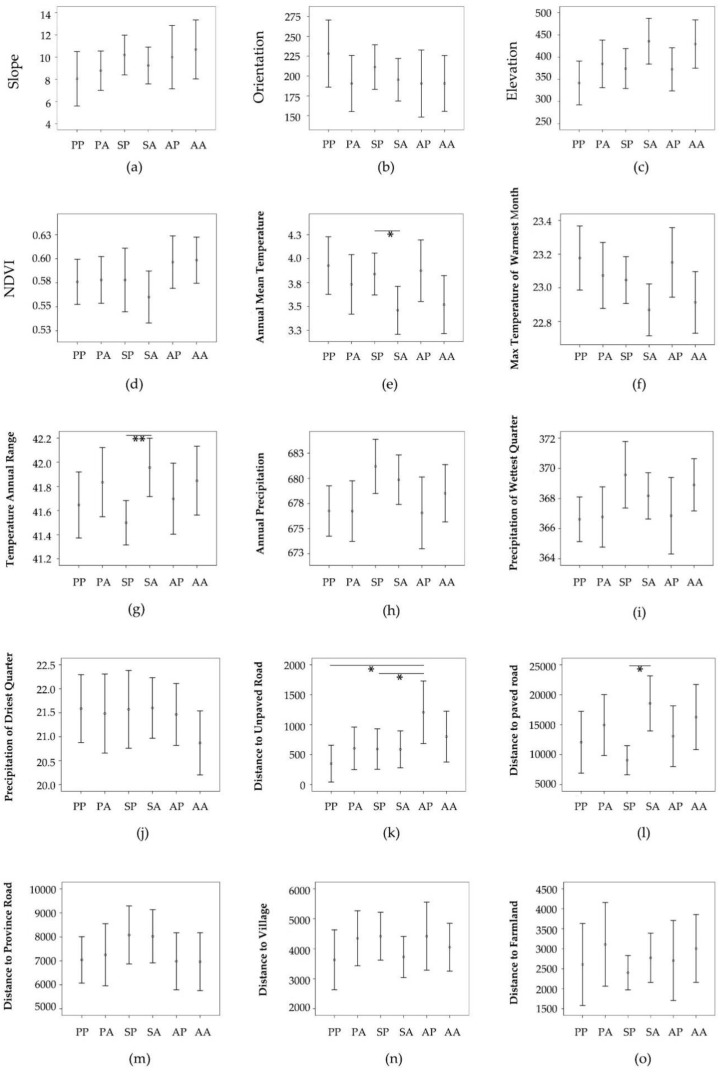
Differences in each environmental variable between presence and absence recordings in each season and differences in topography, vegetation, and human disturbances of presence recordings among three different seasons. The subfigures (**a**–**o**) are the 15 environmental variables from Slope to Distance to farmland excluding Vegetation type (Table 1). PP, spring presence recordings; SP, summer presence recordings; AP, autumn presence recordings; PA, spring absence recordings; SA, summer absence recordings; AA, autumn absence recording. Upper and lower edges show upper and lower bounds of the 95% confidence interval, respectively. The central points are means. * *p* < 0.05; ** *p* < 0.01.

**Figure 2 animals-13-02025-f002:**
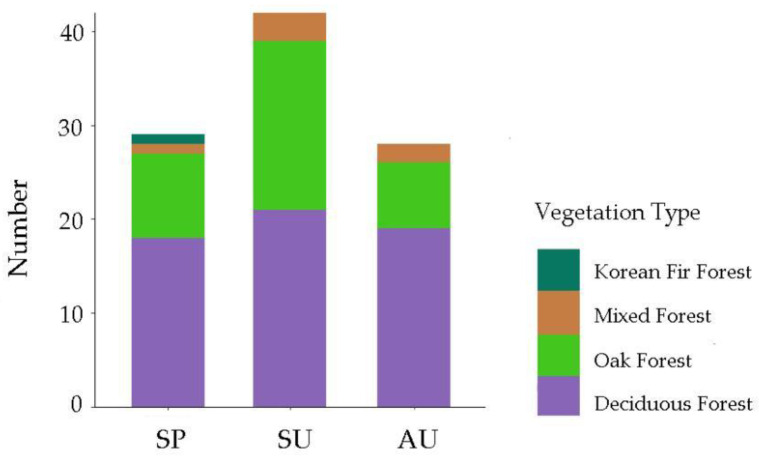
We record the vegetation type for each presence recording, with one vegetation type (whether deciduous, coniferous, or other) for each presence recording, and record the cumulative number of occurrences of each vegetation type in the presence recordings. The vegetation types of presence recordings in each season. SP, spring presence recordings; SU, summer presence recordings; AU, autumn presence recordings; Number, the number of occurrences of each vegetation type photographed.

**Table 1 animals-13-02025-t001:** Environmental variables affecting spatial distribution dynamics of hazel grouse populations.

Type	No.	Variable
Topography	1	Slope
	2	Orientation
	3	Elevation
Vegetation	4	NDVI
	5	Vegetation type
Climate	6	Annual mean temperature
	7	Max. temperature of warmest month
	8	Annual temperature range
	9	Annual precipitation
	10	Precipitation of wettest quarter
	11	Precipitation of driest quarter
Human Disturbance	12	Distance to unpaved road
	13	Distance to paved road
	14	Distance to province road
	15	Distance to village
	16	Distance to farmland

**Table 2 animals-13-02025-t002:** Differences in all variables between presence and absence recordings in each season.

Mann–Whitney U Test									
Environmental Variable	Spring Pr vs. Ab			Summer Pr vs. Ab			Autumn Pr vs. Ab		
	*W*	*t*	*p*	*W*	*t*	*p*	*W*	*t*	*p*
Slope	982.0	0.541	0.589	1919.5	−0.514	0.607	967.0	0.562	0.574
Orientation	821.5	−1.837	0.066	1866.0	−0.969	0.332	910.5	−0.296	0.767
Elevation	1021.5	1.126	0.260	2152.0	1.462	0.144	1026.0	1.459	0.145
NDVI	992.0	0.689	0.491	1842.5	−1.169	0.242	941.0	0.167	0.867
Annual mean temperature	866.5	−1.171	0.242	1744.0	−2.007	**0.045 ***	818.0	−1.702	0.089
Max. temperature of warmest month	894.0	−0.763	0.445	1806.5	−1.476	0.140	811.0	−1.808	0.071
Annual temperature range	1016.0	1.045	0.296	2292.0	2.654	**0.008 ****	984.0	0.821	0.412
Annual precipitation	941.0	−0.067	0.947	1857.0	−1.049	0.294	999.5	1.058	0.290
Precipitation of wettest quarter	963.5	0.269	0.788	1916.0	−0.549	0.583	1035.0	1.614	0.106
Precipitation of driest quarter	920.5	−0.382	0.703	1904.5	−0.659	0.510	841.5	−1.372	0.170
Distance to unpaved road	970.5	0.393	0.694	1910.5	−0.640	0.522	846.5	−1.285	0.199
Distance to paved road	1047.5	1.511	0.131	2232.0	2.142	**0.032 ***	977.5	0.722	0.471
Distance to province road	936.0	−0.141	0.888	2039.0	0.506	0.613	918.5	−0.175	0.861
Distance to village	1021.0	1.118	0.264	1845.5	−1.143	0.253	897.0	−0.501	0.616
Distance to farmland	987.0	0.615	0.539	1994.0	0.119	0.905	979.0	0.744	0.457
**Correspondence Analysis**									
**Environmental Variable**	**Season**			** *χ^2^* **		** *df* **		** *p* **	
Vegetation type	Spring Pr vs. Ab			1.349		8		0.995	
	Summer Pr vs. Ab			6.735		8		0.565	
	Autumn Pr vs. Ab			2.855		8		0.943	

* *p* < 0.05; ** *p* < 0.01. Pr, presence recording; Ab, absence recording. Results with significant differences are shown in bold.

**Table 3 animals-13-02025-t003:** Differences in topography, vegetation, and human disturbances in presence recordings among three different seasons.

Kruskal–Wallis Test			
Environmental Variable	*t*	*df*	*p*
Slope	1.245	2	0.537
Orientation	2.493	2	0.288
Elevation	1.115	2	0.573
Distance to unpaved road	8.650	2	**0.013 ***
Distance to paved road	0.474	2	0.789
Distance to province road	0.374	2	0.829
Distance to village	1.447	2	0.485
Distance to farmland	0.146	2	0.930
NDVI	2.270	2	0.321
**Correspondence Analysis**			
**Environmental Variable**	** *χ* ^2^ **	** *df* **	** *p* **
Vegetation type	5.566	16	0.992

* *p* < 0.05. Results with significant differences are shown in bold.

## Data Availability

The data are not publicly available due to privacy and ethical restrictions.

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
