# Peer review of "Decomposing the Spatial and Temporal Effects of Climate and Habitat on a Hazel Grouse (Tetrastes bonasia) Population in Northeastern Chinese Mountains"

_animals, 2023, doi:10.3390/ani13122025_

Round 1
Reviewer 1 Report
I have read the manuscript of Xing et al. on the Hazel grouse in NE China.
Unfortunately, the English is sometimes very difficult to understand, and everywhere is not fluent at all. I believe that the manuscript absolutely needs the revision by a native English speaker.
Here I reported some language corrections, but they are not all the corrections that are necessary, on the contrary, they are just a sample of corrections that have to be done.
On the attached PDF file I listed other suggestions or requests for changes.

Difficult to understand, and everywhere is not fluent at all.
Author Response
Reviewer 1:
I have read the manuscript of Xing et al. on the Hazel grouse in NE China.
Unfortunately, the English is sometimes very difficult to understand, and everywhere is not fluent at all. I believe that the manuscript absolutely needs the revision by a native English speaker.
Here I reported some language corrections, but they are not all the corrections that are necessary, on the contrary, they are just a sample of corrections that have to be done.
On the attached PDF file I listed other suggestions or requests for changes.
Reply: Thanks for the helpful and constructive comments on this manuscript!
The English has been edited and improved. For your questions about the results in the climate section of the results, our results did show this. In summer, compared to grouse-absence sites, presence sites have higher Annual Mean Temperature, and lower Temperature Annual Range, but there is no significant difference in Max Temperature of Warmest Month.
Reviewer 2 Report
Review for Animals Manuscript ID: animals-2397813
Article: Decomposing the spatial and temporal effects of climate and habitat on Hazel Grouse population in northeastern Chinese mountains
Comments to the Author
This paper deals with a subject that would be of interest to readers of Animals and deals with a subject that excites controversy within the Human-animal conflicts. It is also a very interesting topic as it transfers the scales to be used to more manageable tools for making management decisions. It is a paper that combines many variables to obtain a management plan for a specific species, Hazel Grouse, but which marks a starting point for future work on the conservation of other species.
In short, this is a nice paper. I think that is an interesting and useful contribution to the literature about conservation ecology and specifically management of endangered species and habitat selection and response to climate change and human disturbance. In general, the writing is clear. The arguments are clearly presented, the results are interesting, and the interpretation of the results justified.
Minor comments:
Line 2: I am not sure if it is due to the publication rules of this journal, but why is the word Nilgai written in capital letters? Common names are usually written in lower case. Why the scientific name is not indicated?
Line 35: Perhaps it is clearer if it is indicated that the animal is a bird. For example, "Hazel Grouse (Tetrastes bonasia) is a bird that typically occurs...".
Line 104-106: It is not clear to me why the predictions are as stated by the authors. Is it intuition? Or is it what theoretically would be what this species would prefer.
Line 121: Could you please mention the camera model and its technical specifications?
Author Response
Reviewer 2:
Minor comments:
Line 2: I am not sure if it is due to the publication rules of this journal, but why is the word Nilgai written in capital letters? Common names are usually written in lower case. Why the scientific name is not indicated?
Reply: We changed the title of the article to: Decomposing the spatial and temporal effects of climate and habitat on hazel grouse (Tetrastes bonasia) population in northeastern Chinese mountains. It’s on line 2.
Line 35: Perhaps it is clearer if it is indicated that the animal is a bird. For example, "Hazel Grouse (Tetrastes bonasia) is a bird that typically occurs...".
Reply: We have changed it to “Hazel Grouse (Tetrastes bonasia) is a bird that typically occurs …”. It’s on line 32.
Line 104-106: It is not clear to me why the predictions are as stated by the authors. Is it intuition? Or is it what theoretically would be what this species would prefer.
Reply: The predictions are based on the studies in other regions or on other grouses which cited in the introduction part. It’s on line 102.
Line 121: Could you please mention the camera model and its technical specifications?
Reply: The cameras used were Ltl6210mc (Ltl Acorn, Zhuhai, China), L710-940 (Yianws, Shenzhen, China), and UVL4 (UOVISION, Shenzhen, China). These cameras can take record photographs and videos in visible light during the day and in infrared at night. It’s on lines 119-122.
Reviewer 3 Report
The authors used IR cameras to study the distribution and habitat use of the hazel grouse in a national park in China. The study has several major flaws. First, the English is very bad and at times almost impossible to understand. I understand this is not the author’s first language, yet they could have done a better job by using Google Translate. Basically each sentence needs careful editing of the English, I tried to give comments where appropriate, however this is a huge task and it is not my job as a reviewer. Secondly, the wrong post-hoc tests were used, so any significant difference between treatments must be double checked. Thirdly, there are clearly not major differences in almost all variables analysed, yet in the discussion the authors boldly claim that their study shows clear significant differences in human disturbance, topography and climate. This must be toned down and carefully edited. Overall, I think that even being an interesting natural note, this is a minor study providing dubious results with bold claims and most of all presented in an almost unreadable English. I therefore recommend rejection in its present form. I hope the comments below will help the authors in preparing a better manuscript.
Lines 9-10: grammatically awkward sentence. It is “The timely monitoring of population fluctuations of species…”. And “discovering causes…” of what? Also, what does the sentence actually means? Please correct.
Line 13: you just mentioned in the same sentence that you are working with the Hazel Grouse, no need to repeat.
Line 13-14: it is “We found that the Hazel Grouse prefers stable climate and avoids local human disturbance…”
Lines 9-19: the whole summary is badly written. Please use a scientific translator, I am sure that even Google Translate would return sentences with a better grammar that what is presented now!!
Line 20: what do you mean “..especially those are being threaten?” It means nothing…
Line 21: well, clearly, but is essential for who? And why? Again, the sentence means nothing if presented like this.
Line 24: it is “human disturbance”, the plural is not necessary
Line 24: What do you mean with “climatic, habitat and human activities”? There is no such a thing as a “climatic” or “habitat” activity. Please correct.
Line 26: what do you mean with a “climatic disturbance”? What is this? Climatic factors are just part of environmental variation, unless you are referring to major climate events like hurricanes, cyclones, tsunamis and so on.
Line 27: besides what?
Line 28-31: awkward grammar…
Lines 35-37: awkward grammar, please correct
Lines 43-45: awkward grammar, please correct
Lines 46-56: basically, each sentence in this paragraph is badly written. Please check the grammar. As a reviewer I expect to be able to understand what the authors are talking about!!!
Line 57: delete the 2nd “factor”
Lines 59-60 what’s the difference between “slope direction” and “slope”?
Line 70: change with “other grouses”
Line 72: change with “precipitation during the breeding season”
Lines 76-78: please correct, I assume you mean that burrowing in snow reduces heat loss and at the same time hides from predators…
Line 86: in what sense “realize”? Perhaps you mean “to ensure stable reproduction”?
Line 86-90: very awkward grammar, please rephrase
Line 91: IR cameras are not used for “protecting”. Rephrase
Line 95: “nimble”???
Line 95-108: there are way too many grammar mistakes and awkward phrasing in these sentences to count! Please check grammar and rephrase!
Line 115-116: what does it mean that “the seasonal climate difference is obvious”? Obvious to whom? Why is it obvious? I don’t even understand what you are referring to…
Line 120-126: which cameras did you use (brand and model)? I still do not understand, where these normal trail cameras with IR light or rather cameras recording only in IR (i.e. no visible light, only recording at night)?
Line 126: what do you mean “debugging the camera”? Debug of what?
Line 129-132: makes no sense listening all these info without knowing which camera was used. The tenses are all over the place, once the present is used, another the future. You must always use past tense in the methods!!
Line 136-138: how do you know that it was the same individual in the photos? Unless these are marked individuals there is no way of knowing if you have the same individual in multiple photos. Clarify.
Line 138-140; I have no idea what you mean here. How can you choose the “same number of camera sites”? Correct the grammar and rephrase.
Line 142: how did you measure slope? And would be more appropriate to rename “aspect” with “orientation”.
Line 152-154: what is the point of defining seasons using the already know definition of season? I do not understand why you need this…
Table 1: the variable 12 should be “distance to unpaved road” and 13 “distance to paved road”. Also, I do not understand the labels BIO1, 5 , 7 etc, where do these come from?
Line 162: “abnormally”??? You mean “not normal” perhaps….
Line 164-165: you do not need to write “box-plots were drawn” or “bar-charts were drawn”….
Line 166: Tukey HSD is a parametric test for pairwise comparisons, so it should not be used after a Kruskal-Wallis with non-normally distributed data. Rather you should use non-parametric alternatives like Dunn’s test.
Line 168: I don’t understand, what does it mean that “vegetation type variable were analysed by corresponding analyses”? Which corresponding analyses?
Lines 170-181: Almost each sentence in this paragraph is hard to understand and with awkward grammar, please correct the English!! On a more methodological note, how come 46 cameras with photos but 101 occupancy sites? Also, you should define in the methods what you mean with occupancy: just detection of the grouse, particular behaviours or else? For me this is just the camera detecting a grouse passing by, how can you be sure that the grouse was actually occupying the area? Finally, you claim to have found climate and human disturbance effects, but present no test results or figures to show this.
Figure 1. I would rather use means and 95% CI, so that real differences are easily visible. Looking at the boxplot it looks like that there is no real difference among any of the occupancy-vacancy comparisons and that any small difference might be the results of using the wrong post-hoc test.
Line 220-223: how can you claim this? You basically found that none of the environmental variables matter, as the miniscule difference of mean summer temperature does not suggest anything at all. The only effect I see (which again is miniscule) is the distance from paved road in summer, which makes sense as the grouse would be in the reproductive period. It is very little to support your claims!
Line 226: grouses forage on the ground, not in the tree canopies. Please, correct
Lines 229-239: but how were the camera distributed in the study area? Mostly in forest (as this was the main vegetation) or actually evenly spread to account for all possible vegetation types? In the first case this of course would bias your result, in the second case you should use some sort of weighted occurrence score. So, I am not sure how can you exactly claim that “topography and vegetation do not limit distribution”.
Lines 240-243: but in your results you found no difference in elevation between occupied and vacant sites (Figure 1), so how can you claim that elevation was lower in occupied sites?
Lines 249-266: the English is really bad in this whole section, please correct the grammar and rephrase.
Lines 268-272: these sentences belong to the introduction, not discussion
Lines 273-279: before claiming this I would suggest to run pairwise comparisons with non-parametric tests.
Lines 280-287: bad English, please check the grammar
Line 288: awkward title, perhaps change with “Human disturbance”
Lines 289-295: awkward English, please correct the grammar
Line 296-297: the sentence makes no sense, please correct
Lines 298-305: awkward English, please check the grammar
Line 306-316: very bad English, almost impossible to understand the meaning. Please rephrase and correct the grammar
Line 322-324: impossible to understand, please rephrase
Line 325: negative what?
Line 329: which areas? The ending sentence is impossible to understand, please rephrase
Line 332: actually the study showed that there is no effect at all of topography or vegetation and only a minor (perhaps negligible) effect of temperature and human disturbance. So, the sentence should be rephrased with “This study shown NO influence of….”
Line 334-335: again, bold claim as the results are weak and the tests used are wrong
As I already mentioned, the quality of the English is terrible. I absolutely understand that this is not the author's first language (it is not mine either!!), however I expect a much better English from researchers wanting to publish in an international journal. In my opinion, it is very disrespectful to submit a manuscript in such a poor shape and expect that a reviewer should 1) understand the meaning of obscure sentences and 2) correct the poor English, grammar mistakes and full meaning of the sentences. This is either a job for a proper translator or for a co-author, not for a reviewer!
Author Response
Reviewer 3:
The authors used IR cameras to study the distribution and habitat use of the hazel grouse in a national park in China. The study has several major flaws. First, the English is very bad and at times almost impossible to understand. I understand this is not the author’s first language, yet they could have done a better job by using Google Translate. Basically each sentence needs careful editing of the English, I tried to give comments where appropriate, however this is a huge task and it is not my job as a reviewer. Secondly, the wrong post-hoc tests were used, so any significant difference between treatments must be double checked. Thirdly, there are clearly not major differences in almost all variables analysed, yet in the discussion the authors boldly claim that their study shows clear significant differences in human disturbance, topography and climate. This must be toned down and carefully edited. Overall, I think that even being an interesting natural note, this is a minor study providing dubious results with bold claims and most of all presented in an almost unreadable English. I therefore recommend rejection in its present form. I hope the comments below will help the authors in preparing a better manuscript.
Lines 9-10: grammatically awkward sentence. It is “The timely monitoring of population fluctuations of species…”. And “discovering causes…” of what? Also, what does the sentence actually means? Please correct.
Reply: Thanks very much for the helpful and constructive comments! We have changed it to “The timely monitoring of population fluctuations of endangered species and discovering its causes are critical for biodiversity conservation in mountainous areas”. It’s on lines 9-10.
Line 13: you just mentioned in the same sentence that you are working with the Hazel Grouse, no need to repeat.
Reply: We have deleted it. It’s on line 13.
Line 13-14: it is “We found that the Hazel Grouse prefers stable climate and avoids local human disturbance…”
Reply: We have changed it to “The hazel grouse preferred stable climate conditions”. It’s on line 13.
Lines 9-19: the whole summary is badly written. Please use a scientific translator, I am sure that even Google Translate would return sentences with a better grammar that what is presented now!!
Reply: The English has been edited and improved by language company.
Line 20: what do you mean “..especially those are being threaten?” It means nothing…
Reply: We have changed it with “and these are especially critical for threatened species”. It’s on line 20.
Line 21: well, clearly, but is essential for who? And why? Again, the sentence means nothing if presented like this.
Reply: We have deleted the whole sentence. It’s on line 20.
Line 24: it is “human disturbance”, the plural is not necessary
Reply: We have changed it to “human disturbance”. It’s on line 22.
Line 24: What do you mean with “climatic, habitat and human activities”? There is no such a thing as a “climatic” or “habitat” activity. Please correct.
Reply: We have deleted them and replaced them with “environment variables”. It’s on line 23.
Line 26: what do you mean with a “climatic disturbance”? What is this? Climatic factors are just part of environmental variation, unless you are referring to major climate events like hurricanes, cyclones, tsunamis and so on.
Reply: We changed it with “Temperatures and roads influenced the distribution of…”, we used “temperature” to replace “climate”. It’s on line 24.
Line 27: besides what?
Reply: We have deleted “Besides”. It’s on line 25.
Line 28-31: awkward grammar…
Reply: The English has been edited and improved by language company.
.
Lines 35-37: awkward grammar, please correct
Reply: The English has been edited and improved by language company.
Lines 43-45: awkward grammar, please correct
Reply: The English has been edited and improved by language company.
Lines 46-56: basically, each sentence in this paragraph is badly written. Please check the grammar. As a reviewer I expect to be able to understand what the authors are talking about!!!
Reply: The English has been edited and improved by language company.
Line 57: delete the 2nd “factor”
Reply: We have deleted the 2nd “factor”. It’s on line 53
Lines 59-60 what’s the difference between “slope direction” and “slope”?
Reply: We have replaced “slope direction” with “orientation”. It’s on line 55.
Line 70: change with “other grouses”
Reply: We have changed it to “other grouses”. It’s on line 69.
Line 72: change with “precipitation during the breeding season”
Reply: We have changed it to “precipitation during the breeding season”. It’s on line 71.
Lines 76-78: please correct, I assume you mean that burrowing in snow reduces heat loss and at the same time hides from predators…
Reply: We have changed it with “…and burrowing in snow reduces heat loss and at the same time hides from predators”. It’s on line 76.
Line 86: in what sense “realize”? Perhaps you mean “to ensure stable reproduction”?
Reply: We have changed it with “to ensure their stable reproduction”. It’s on line 84.
Line 86-90: very awkward grammar, please rephrase
Reply: The English has been edited and improved by language company.
Line 91: IR cameras are not used for “protecting”. Rephrase
Reply: We have deleted it. It’s on line 90.
Line 95: “nimble”???
Reply: We have replaced it with “fast-moving”. It’s on line 94.
Line 95-108: there are way too many grammar mistakes and awkward phrasing in these sentences to count! Please check grammar and rephrase!
Reply: The English has been edited and improved by language company.
Line 115-116: what does it mean that “the seasonal climate difference is obvious”? Obvious to whom? Why is it obvious? I don’t even understand what you are referring to…
Reply: What we mean is that the climate varies greatly between seasons. It’s on lines 114-115.
Line 120-126: which cameras did you use (brand and model)? I still do not understand, where these normal trail cameras with IR light or rather cameras recording only in IR (i.e. no visible light, only recording at night)?
Reply: The cameras used were Ltl6210mc (Ltl Acorn, Zhuhai, China), L710-940 (Yianws, Shenzhen, China), and UVL4 (UOVISION, Shenzhen, China). These cameras can take record photographs and videos in visible light during the day and in infrared at night. It’s on lines 119-122.
Line 126: what do you mean “debugging the camera”? Debug of what?
Reply: We mean the camera view and shooting angle were estimated and adjusted, and the appropriate lens direction was selected to obtain the best possible view. It’s on lines 127-129.
Line 129-132: makes no sense listening all these info without knowing which camera was used. The tenses are all over the place, once the present is used, another the future. You must always use past tense in the methods!!
Reply: The cameras used were Ltl6210mc (Ltl Acorn, Zhuhai, China), L710-940 (Yianws, Shenzhen, China), and UVL4 (UOVISION, Shenzhen, China). These cameras can take record photographs and videos in visible light during the day and in infrared at night. It’s on lines 119-122. The English has been edited and improved by language company.
Line 136-138: how do you know that it was the same individual in the photos? Unless these are marked individuals there is no way of knowing if you have the same individual in multiple photos. Clarify.
Reply: We considered it as the same individual, because it kept appearing in a continuous series of photos or videos and moving continuously in space. It’s on lines 140-142.
Line 138-140; I have no idea what you mean here. How can you choose the “same number of camera sites”? Correct the grammar and rephrase.
Reply: The English has been edited and improved by language company.
Continuous photos or videos of the same individual at the same camera site within 30 min were counted as one independent valid photo. We considered it as the same individual, because it kept appearing in a continuous series of photos or videos and moving continuously in space. For the cameras that captured the hazel grouse, we defined each valid photo of the hazel grouse as a presence site. This means that the same camera site could represent many presence sites at different times. And we defined the presence just detection of the grouse. For the cameras that did not capture the hazel grouse but captured other animals, the same method was used, and each valid photo of an animal was recorded as an absence site. The number of presence sites and the number of absence sites were counted, and we randomly selected the same number (as presence sites) from the absence sites for comparison and analysis. It’s on lines 139-151.
Line 142: how did you measure slope? And would be more appropriate to rename “aspect” with “orientation”.
Reply: It was assigned by importing the Digital Elevation Model of Hunchun area into Arcgis, and we have replaced “aspect” with “orientation”. It’s on lines 154-155.
Line 152-154: what is the point of defining seasons using the already know definition of season? I do not understand why you need this…
Reply: We have deleted it.
Table 1: the variable 12 should be “distance to unpaved road” and 13 “distance to paved road”. Also, I do not understand the labels BIO1, 5, 7 etc, where do these come from?
Reply: We have used “distance to unpaved road” and “distance to paved road” to replace “distance to mud road” and “distance to concrete road” and we have deleted BIO1, 5, 7, 12, 16 and 17. It’s on Table 1.
Line 162: “abnormally”??? You mean “not normal” perhaps….
Reply: We have replaced it with “not normal”. It’s on line 173.
Line 164-165: you do not need to write “box-plots were drawn” or “bar-charts were drawn”….
Reply: We have deleted them. It’s on line 177.
Line 166: Tukey HSD is a parametric test for pairwise comparisons, so it should not be used after a Kruskal-Wallis with non-normally distributed data. Rather you should use non-parametric alternatives like Dunn’s test.
Reply: There was a mistake in previous statement. We used Kruskal-Wallis test with Bonferroni correction and Mann-Whitney U test rather than Tukey HSD, all the analyses we used were non-parametric tests. It’s on lines 177-178.
Line 168: I don’t understand, what does it mean that “vegetation type variable were analysed by corresponding analyses”? Which corresponding analyses?
Reply: We used Correspondence analysis (R-Q factor analysis) to explore whether Vegetation Type variable between presence and absence sites in each season or differed across seasons at the presence sites. It’s on lines 175-177, 180-181, Table 2,3 and Figure 2.
Lines 170-181: Almost each sentence in this paragraph is hard to understand and with awkward grammar, please correct the English!! On a more methodological note, how come 46 cameras with photos but 101 occupancy sites? Also, you should define in the methods what you mean with occupancy: just detection of the grouse, particular behaviours or else? For me this is just the camera detecting a grouse passing by, how can you be sure that the grouse was actually occupying the area? Finally, you claim to have found climate and human disturbance effects, but present no test results or figures to show this.
Reply: The English has been edited and improved by language company.
Continuous photos or videos of the same individual at the same camera site within 30 min were counted as one independent valid photo. We considered it as the same individual, because it kept appearing in a continuous series of photos or videos and moving continuously in space. For the cameras that captured the hazel grouse, we defined each valid photo of the hazel grouse as a presence site. This means that the same camera site could represent many presence sites at different times. And we defined the presence just detection of the grouse. In this study we only explored where the Hazel grouse prefer to move around, but did not further explore whether the Hazel grouse occupy these places. We deleted the bold claims. It’s on lines 139-145, 190.
Figure 1. I would rather use means and 95% CI, so that real differences are easily visible. Looking at the boxplot it looks like that there is no real difference among any of the occupancy-vacancy comparisons and that any small difference might be the results of using the wrong post-hoc test.
Reply: We replotted the figure with means and 95% CI. It’s on Figure 1.
Line 220-223: how can you claim this? You basically found that none of the environmental variables matter, as the miniscule difference of mean summer temperature does not suggest anything at all. The only effect I see (which again is miniscule) is the distance from paved road in summer, which makes sense as the grouse would be in the reproductive period. It is very little to support your claims!
Reply: We deleted the bold claims. It’s on lines 226-227.
Line 226: grouses forage on the ground, not in the tree canopies. Please, correct
Reply: We have changed it with “because they hide in snow burrows in winter”. It’s on lines 228-229.
Lines 229-239: but how were the camera distributed in the study area? Mostly in forest (as this was the main vegetation) or actually evenly spread to account for all possible vegetation types? In the first case this of course would bias your result, in the second case you should use some sort of weighted occurrence score. So, I am not sure how can you exactly claim that “topography and vegetation do not limit distribution”.
Reply: Most of cameras were located in the forest, so this may have introduced bias into the data. These results cannot indicate a habitat preference of the hazel grouse and further exploration is needed. And we have deleted the bold claims. It’s on lines 239-240.
Lines 240-243: but in your results you found no difference in elevation between occupied and vacant sites (Figure 1), so how can you claim that elevation was lower in occupied sites?
Reply: We deleted the whole sentence. It’s on line 241.
Lines 249-266: the English is really bad in this whole section, please correct the grammar and rephrase.
Reply: The English has been edited and improved by language company.
Lines 268-272: these sentences belong to the introduction, not discussion
Reply: We have moved these sentences to the introduction. It’s on lines 63-66.
Lines 273-279: before claiming this I would suggest to run pairwise comparisons with non-parametric tests.
Reply: We made a mistake in our previous statement. We used Kruskal-Wallis test with Bonferroni correction and Mann-Whitney U test rather than Tukey HSD, all the analyses we used were non-parametric tests. It’s on lines 177-178.
Lines 280-287: bad English, please check the grammar
Reply: The English has been edited and improved by language company.
Line 288: awkward title, perhaps change with “Human disturbance”
Reply: We have changed the title with “Human disturbance”. It’s on line 283.
Lines 289-295: awkward English, please correct the grammar
Reply: The English has been edited and improved by language company.
Line 296-297: the sentence makes no sense, please correct
Reply: We changed it with “Food resources become plentiful in multiple areas in the summer, but thick vegetation also increases predation risk.” It’s on lines 290-291.
Lines 298-305: awkward English, please check the grammar
Reply: The English has been edited and improved by language company.
Line 306-316: very bad English, almost impossible to understand the meaning. Please rephrase and correct the grammar
Reply: The English has been edited and improved by language company.
Line 322-324: impossible to understand, please rephrase
Reply: The English has been edited and improved by language company.
Line 325: negative what?
Reply: What we mean is that human activities can negatively affect the Hazel Grouse by forcing them to move away from unpaved roads. It’s on lines 319-320.
Line 329: which areas? The ending sentence is impossible to understand, please rephrase
Reply: We mean the core of the nature reserve. It’s on line 322.
Line 332: actually the study showed that there is no effect at all of topography or vegetation and only a minor (perhaps negligible) effect of temperature and human disturbance. So, the sentence should be rephrased with “This study shown NO influence of….”
Reply: We changed it to “This study showed no influence of topography and vegetation, while temperature and human disturbance can affect the population distribution…”. It’s on lines 326-328.
Line 334-335: again, bold claim as the results are weak and the tests used are wrong
Reply: We changed it to “The distribution of hazel grouse in the park was affected by temperature and roads” It’s on line 330.
Round 2
Reviewer 1 Report
The Authors addressed all requests for corrections. The text is greatly improved, I suggest publishing the manuscript.
Author Response
We have no response because there were no comments or suggestions for revise.
Reviewer 3 Report
I am glad to see a massive improvement of the English, the manuscript is now much easier to read and understand. There are still some awkward sentences and grammar mistakes here and there, I marked some. However, the manuscripts still requires some major revisions, particularly in the methods and result sections, see my comments below.
Lines 26-27: I would cut everything before the comma and start with “This study provides ecological information….”
Lines 97-106: These whole section is written to go either in results or discussion. This is the introduction, so you should state why the study was done and what you expect, not what you found.
Line 111: the altitude of the park seems excessive, is this really 5973m?
Line 120: delete “take”
Line 129: change with “three photos plus 30s videos after triggering”
Line 140-141: but then the presence site was considered multiple times within the same year? Or between years? To me “presence site” is simply the number of valid photos (with grouse) you collected over time, so it is still unclear to me why you call it “site”. Also, change with “We defined presence as the detection of a grouse by the trail cameras”.
Lines 141-143: what do you mean with valid photo of an animal? Which animal? You mean not a grouse? You never had a misfire by the cameras in 9 years?
Lines 144-146: what do you mean you “randomly selected the same number”, how did you choose the sites to analyse? Was this a blind choice by someone else, a random draw or something else? And what this random also across years?
Table 1: you already defined in the text the acronym NDVI, so you can use this in the table as well. Why only max temperature, what about minimum temperature? Also, was max temperature only in the warmest month on average or just the max temperature recorded over the entire year? Also, in the section “Climate” it should be clarified if these are annual values, otherwise it looks like these are single values averaged over 9 years of monitoring
Line 173-174: isn’t this a repetition of line 170? Or perhaps is this referring to the analysis of only presence sites? Please clarify
Line 179: to me the usage of “site” is confusing. You write that “46 IR cameras had images” but “101 sites were counted”. How can this be? For me should be rather that “46 sites showed presence of the grouse with 101 recordings (video-photos) of behavioural activity”. This means that in 46 2x2 km squares sites you did find presence of the grouse (because the cameras recorded it) and in total you have 101 actual recordings. If you use the same cameras multiple times without accounting for it in the statistics you commit pseudoreplication and inflate the occurrence of the grouse in your study area. Please clarify.
Table 2: I am pretty confident there is better way to present the results of this table without wasting 3 pages!
Line 187: why “winter excluded”? You do not provide any rationale in the methods for presenting a 3-seasons result. Please clarify.
Line 205: what do you mean “counted the number of each vegetation type”? The number of what: tree, bushes, saplings? Or you mean the relative density of each vegetation type in each plot?
Line 207-208: photographed where? In the camera recordings? But then how did you estimate the “number”? Please explain this in the methods and clarify.
Line 221-222: so this should be explained in the methods and clarify why winter was not included in the analysis, i.e. simply there is not enough grouse activity.
Line 228: I strongly disagree that plain areas ARE for human activity, they simply are more commonly exploited by humans
Line 240-241: where did you state this hypothesis? I can’t find it in the introduction, which is where it should be.
Lines 243-244: there is no mention of how large was the monitored area. Anyway, I doubt that the authors covered so many different types of forests in their sampling. Rather, what you sampled are areas with different tree communities within the park which main habitat is a forest mosaic. So, your camera trapping is only telling that deciduous patches/areas (not forests) are preferentially used than coniferous ones. Same for the rest of the paragraph, these are not “forests” rather “patches” or “areas” with a dominant vegetation type.
Lines 256-258: please correct the English here
Line 268-271: but how do you know that they prefer breeding in such areas? You only used video monitoring and have no way to tell if the birds were actually breeding there or not, unless you have clear evidence of that, for example found the nests or recorded females with the brood. It might simply be that these are the best areas to search for food. Or, grouse were simply spending some time in cooler areas as the temperature range is quite high (>40 C) and the presence is higher in lower temperatures. Essentially, with the data you present it is impossible to say anything about breeding.
Line 280-281: the park is forest, the oak vegetation type is more common at the edges of such forest.
Line 283: how can thick vegetation increase predation risk? Should be the opposite as grouse are more concealed from predators in thick vegetation. This is indeed where they make their nests.
Line 284-288: again, do you have any data to support the fact that hens with chicks move closer to roads in spring or summer?
Line 300-301: do young stay with the hens until Autumn?
Lines 308-311: not clear what you mean here, which other side?
Line 312: negative effects such as….???
Lines 315-317: you say that the human activity occurs also in the core of the nature reserve. Perhaps it would be good to include some data about human activity in your field area as you main point is that human disturbance somehow affects the hazel grouse (not clear how). However, it is not clear from your data what is the level of human activity in the park. I would assume that after 9 years of monitoring you should have some evidence of human activity, for example on roads or forest paths.
Line 324: I can see how putting signs along roads can alert people of the presence of grouse, but how can the protection of the grouse focus on temperature?
Line 327: what does it mean “human disturbances also significantly affect species”? Which species? How?
Line 328-329: awkward English, rephrase.
The overall quality of the English language is now acceptable, only minor revisions are needed
Author Response
Reviewer 3:
I am glad to see a massive improvement of the English, the manuscript is now much easier to read and understand. There are still some awkward sentences and grammar mistakes here and there, I marked some. However, the manuscripts still requires some major revisions, particularly in the methods and result sections, see my comments below.
Lines 26-27: I would cut everything before the comma and start with “This study provides ecological information….”
Reply: We have changed it to “This study provides ecological information….”. It’s on lines 34-36.
Lines 97-106: These whole section is written to go either in results or discussion. This is the introduction, so you should state why the study was done and what you expect, not what you found.
Reply: We changed it with “we expected that the hazel grouse prefers…”. It’s on line 138.
Line 111: the altitude of the park seems excessive, is this really 5973m?
Reply: The altitude of the park is 5-973m, we have changed it. It’s on line 150.
Line 129: change with “three photos plus 30s videos after triggering”
Reply: We changed it with “three photos plus 30s videos after triggering”. It’s on lines 172-173.
Line 140-141: but then the presence site was considered multiple times within the same year? Or between years? To me “presence site” is simply the number of valid photos (with grouse) you collected over time, so it is still unclear to me why you call it “site”. Also, change with “We defined presence as the detection of a grouse by the trail cameras”.
Reply: A camera site may have many presence recordings, but each presence recording was independent. We changed “presence site” with “presence recording”, and changed “absence site” with “absence recording”. We changed it with “We defined presence as the detection of a grouse by the trail cameras”. It’s on lines 192-193.
Lines 141-143: what do you mean with valid photo of an animal? Which animal? You mean not a grouse? You never had a misfire by the cameras in 9 years?
Reply: We mean any birds except hazel grouse. Once we got a valid photo of a bird, we collected the data according to the time and place of its occurrence and classified it according to the season of its presence, with data for each season being independent. For example, if a particular camera did not record a hazel grouse throughout the spring over all the 10-year period, then it was considered that no hazel grouse was present at that camera site in the spring, and the recordings of other birds it captured were treated as absence recordings for the spring. If, however, the site has photographed a hazel grouse in summer, then the valid photo of the hazel grouse can be considered as a summer presence recording, and the other birds photographed by this camera in summer cannot be considered as absence recordings. It’s on lines 193-203.
Lines 144-146: what do you mean you “randomly selected the same number”, how did you choose the sites to analyse? Was this a blind choice by someone else, a random draw or something else? And what this random also across years?
Reply: We selected a certain number of recordings from the absence recordings with simple random sampling to compare with the presence recordings for analysis. We did not consider inter-annual differences. It’s on lines 205-206.
Table 1: you already defined in the text the acronym NDVI, so you can use this in the table as well. Why only max temperature, what about minimum temperature? Also, was max temperature only in the warmest month on average or just the max temperature recorded over the entire year? Also, in the section “Climate” it should be clarified if these are annual values, otherwise it looks like these are single values averaged over 9 years of monitoring
Reply: We changed it with “NDVI”. There was only max temperature because there was something wrong with the minimum temperature of coldest month, it was not accurate so we did not use it. The max temperature of warmest month was the max temperature of warmest month recorded over the entire year. Clarification is now made that all these are annual values on lines 220-221.
Line 173-174: isn’t this a repetition of line 170? Or perhaps is this referring to the analysis of only presence sites? Please clarify
Reply: It was not a repetition. We used the correspondence analysis to compare vegetation type between presence and absence recordings in each season on lines 236-237. And we used the correspondence analysis to compare vegetation type between presence recordings from different seasons on lines 242-243.
Line 179: to me the usage of “site” is confusing. You write that “46 IR cameras had images” but “101 sites were counted”. How can this be? For me should be rather that “46 sites showed presence of the grouse with 101 recordings (video-photos) of behavioural activity”. This means that in 46 2x2 km squares sites you did find presence of the grouse (because the cameras recorded it) and in total you have 101 actual recordings. If you use the same cameras multiple times without accounting for it in the statistics you commit pseudoreplication and inflate the occurrence of the grouse in your study area. Please clarify.
Reply: We changed “presence site” with “presence recording”, and changed “absence site” with “absence recording”. There do exist circumstances, in which the same cameras recorded multiple presence recordings, but the recordings of the same camera were on different time and date. It’s on lines 191-192.
Table 2: I am pretty confident there is better way to present the results of this table without wasting 3 pages!
Reply: We have simplified the table. It’s on Table 2.
Line 187: why “winter excluded”? You do not provide any rationale in the methods for presenting a 3-seasons result. Please clarify.
Reply: Because there were only two winter presence recordings, we did not analyze variables of presence and absence sites in winter. It’s on lines 252-255.
Line 205: what do you mean “counted the number of each vegetation type”? The number of what: tree, bushes, saplings? Or you mean the relative density of each vegetation type in each plot?
Reply: We mean to record the vegetation type for each presence recording, with one vegetation type (whether deciduous, coniferous or other) for each presence recording, and to record the cumulative number of occurrences of each vegetation type in the presence recordings. It’s on lines 297-299.
Line 207-208: photographed where? In the camera recordings? But then how did you estimate the “number”? Please explain this in the methods and clarify.
Reply: We mean to record the vegetation type for each presence recording, with one vegetation type (whether deciduous, coniferous or other) for each presence recording, and to record the cumulative number of occurrences of each vegetation type in the presence recordings. It’s on lines 297-299, 303-304.
Line 221-222: so this should be explained in the methods and clarify why winter was not included in the analysis, i.e. simply there is not enough grouse activity.
Reply: There were only two winter presence recordings, we did not analyze variables of presence and absence sites in winter. It’s on lines 252-255.
Line 228: I strongly disagree that plain areas ARE for human activity, they simply are more commonly exploited by humans
Reply: We changed it with “while the plain areas are more commonly exploited by humans”. It’s on lines 334-335.
Line 240-241: where did you state this hypothesis? I can’t find it in the introduction, which is where it should be.
Reply: We changed it with “that hazel grouse prefer distributing in patches dominated by deciduous trees”. It’s on lines 352-353. And we stated this hypothesis on lines 138- 139.
Lines 243-244: there is no mention of how large was the monitored area. Anyway, I doubt that the authors covered so many different types of forests in their sampling. Rather, what you sampled are areas with different tree communities within the park which main habitat is a forest mosaic. So, your camera trapping is only telling that deciduous patches/areas (not forests) are preferentially used than coniferous ones. Same for the rest of the paragraph, these are not “forests” rather “patches” or “areas” with a dominant vegetation type.
Reply: We changed “forests” with “patches”. It’s on lines 357-358.
Lines 256-258: please correct the English here
Reply: We changed it with “Because the winter is a critical period for hazel grouse, factors affecting the winter distribution of the hazel grouse need further research”. It’s on lines 373-376.
Line 268-271: but how do you know that they prefer breeding in such areas? You only used video monitoring and have no way to tell if the birds were actually breeding there or not, unless you have clear evidence of that, for example found the nests or recorded females with the brood. It might simply be that these are the best areas to search for food. Or, grouse were simply spending some time in cooler areas as the temperature range is quite high (>40 C) and the presence is higher in lower temperatures. Essentially, with the data you present it is impossible to say anything about breeding.
Reply: We changed it with “a smaller temperature annual range may indicate that the hazel grouse preferred an environment with stable temperature in breeding season”. It’s on lines 398-399.
Line 280-281: the park is forest, the oak vegetation type is more common at the edges of such forest.
Reply: We have deleted it. It’s on line 411.
Line 283: how can thick vegetation increase predation risk? Should be the opposite as grouse are more concealed from predators in thick vegetation. This is indeed where they make their nests.
Reply: We have deleted it. It’s on line 420.
Line 284-288: again, do you have any data to support the fact that hens with chicks move closer to roads in spring or summer?
Reply: We have deleted it. It’s on line 420.
Line 300-301: do young stay with the hens until Autumn?
Reply: Young birds already live independently in the autumn. We deleted “maybe with chicks”. It’s on line 437.
Lines 308-311: not clear what you mean here, which other side?
Reply: What we mean was that despite the camera's location was closed to the road, most of the local roads were adjacent to mountains and the side close to the road was usually steep and smooth, separating the road from the forest like a barrier. It’s on lines 446-448.
Line 312: negative effects such as….???
Reply: Villagers collecting mushrooms walking along the road made various noises that may scare the grouse. It’s on lines 455-456.
Lines 315-317: you say that the human activity occurs also in the core of the nature reserve. Perhaps it would be good to include some data about human activity in your field area as you main point is that human disturbance somehow affects the hazel grouse (not clear how). However, it is not clear from your data what is the level of human activity in the park. I would assume that after 9 years of monitoring you should have some evidence of human activity, for example on roads or forest paths.
Reply: Through experiments and interactions with local villagers, we found that many human activities still exist deep in the reserve, such as cattle grazing and farmers collecting mushrooms and wild vegetables. It’s on lines 459-461.
Line 324: I can see how putting signs along roads can alert people of the presence of grouse, but how can the protection of the grouse focus on temperature?
Reply: We changed it with “the protection of the hazel grouse in the national park should focus on areas near paved roads in summer, such as putting signs along roads to alert people of the presence of grouse”. It’s on lines 470-473.
Line 327: what does it mean “human disturbances also significantly affect species”? Which species? How?
Reply: We changed it with “Understanding the habitat and climate requirements of endangered species is important for species conservation, and the effect of human disturbance on them also needs to be clarified”. It’s on lines 474-476.
Line 328-329: awkward English, rephrase.
Reply: We changed it with “We need to know more about the habitats where endangered species live to protect them and prevent them from extinction”. It’s on lines 478-480.
